# A Profit Maximization Model for Data Consumers with Data Providers' Incentives in Personal Data Trading Market

Hyojin Park [1,2], Hyeontaek Oh [3,*] and Jun Kyun Choi [4]

1    Department of Information and Communications Engineering, Korea Advanced Institute of Science and Technology (KAIST), Daejeon 34141, Republic of Korea; gaiaphj@kaist.ac.kr
2    TOVDATA Inc., Daejeon 34141, Republic of Korea
3    Institute for Information Technology Convergence, Korea Advanced Institute of Science and Technology (KAIST), Daejeon 34141, Republic of Korea
4    School of Electrical Engineering, Korea Advanced Institute of Science and Technology (KAIST), Daejeon 34141, Republic of Korea; jkchoi59@kaist.ac.kr
*    Correspondence: hyeontaek@kaist.ac.kr

**Abstract:** This paper proposes a profit maximization model for a data consumer when it buys personal data from data providers (by obtaining consent) through data brokers and provides their new services to data providers (i.e., service consumers). To observe the behavioral models of data providers, the data consumer, and service consumers, this paper proposes the willingness-to-sell model of personal data of data providers (which is affected by data providers' behavior related to explicit consent), the service quality model obtained by the collected personal data from the data consumer's perspective, and the willingness-to-pay model of service consumers regarding provided new services from the data consumer. Particularly, this paper jointly considers the behavior of data providers and service users under a limited budget. With parameters inspired by real-world surveys on data providers, this paper shows various numerical results to check the feasibility of the proposed models.

**Keywords:** consent; personal data; data trading; data consumer; profit maximization

## 1. Introduction

Today, the power of big data and data analytic technologies such as artificial intelligence (AI) and machine learning (ML) have been discovered, and having good quality data is important to companies and organizations for increasing their competitive power in business. Particularly, personal data draws huge attention to companies and organizations because the data can be used to extract various information about customers for new value-added services. However, from the data consumer's perspective, it is not easy to directly collect and aggregate good quality data because it is necessary to consider not only the dimensionality/complexity of data having various structured/unstructured formats but also geographical and social issues related data collection. Therefore, to obtain good quality data, the necessity of a data trading market has arisen, and now many data trading transactions are done through data brokers [1–3].

Since personal data have become an important asset, the data broker market size has continuously grown [4], which collects personal data from various sources and sells the data to third parties [5]. Specifically, data brokers collect personal data by providing services with opt-in-based privacy policy agreements and exploit personal data without any additional notice or consent from data subjects (or personal data providers); then, they sell collected personal data to many data consumers. In this process, there was no consideration of direct incentives to the data providers.

Therefore, conventional studies on data trading have mainly considered the behavior of data brokers and data consumers [6–9]. Mainly, to analyze the profit of data brokers, willingness-to-pay (WTP) models of data consumers that utilize the provided dataset

from data brokers have mainly focused on data trading markets. However, such personal data exploitation of the data brokers causes adversarial effects on data subjects such as privacy infringement and alleviated risks of data breaches. Therefore, the necessity of new regulations or guidelines for protecting data subjects' rights while encouraging data utilization has increased [10].

Based on these backgrounds, many countries have established new data regulations (e.g., General Data Protection Regulation (GDPR) in the European Union [11] and California Consumer Privacy Act in the United States [12]) to satisfy both the need to empower data subjects on personal data protection and provide guidelines for personal data utilization. Particularly, since new data-related regulations and governance have emphasized individuals' rights and consent management, companies and organizations now need to obtain explicit consent or agreement from data subjects (or personal data providers) for utilizing personal data. Consequently, the personal data provider becomes an important stakeholder in the personal data ecosystem. Therefore, to support and manage explicit consent for data sharing, the concept of a consent management system has been studied [13,14] that can support personal data trading with explicit consent.

Moreover, the data providers now require direct incentives (e.g., as forms of mileage, cash, cryptocurrency, etc.) for using their personal data. Therefore, there are data brokering services that have focused on direct incentives to data providers (such as Datacoup [15], Mydata Operators [16], etc.) for compensating personal data use. Now, the data consumers need to consider the direct incentives to the data providers as a new cost for utilizing personal data for their new business; in other words, the data consumers need to actually pay for using the personal data of the data providers. An example data trading scenario is as follows: A data provider can post its GPS data collected from a smartwatch to a data broker. A data consumer offers a price for buying GPS data from the data provider. If the data provider accepts the offered price, the data consumer buys the posted GPS data with the explicit consent of the data provider. The data consumer contacts many data providers to buy GPS data for creating new recommendation services based on the users' position.

According to the literature in economics [17–19], people are willing to share their personal data by taking proper amounts of incentives although they think that protecting their privacy is important; moreover, people now know that their personal data have economic values [20–22]. Accordingly, in several studies, data providers' behavior has been considered as willingness-to-share/sell (WTS) personal data models [23–27], which is related to obtaining explicit consent from data providers.

Based on the behavioral models of data consumers and data providers (i.e., WTP and WTS), many studies mentioned above have focused on analyzing the behavior of data brokers (e.g., profit maximization, cost minimization, etc.). The authors' previous studies [25,26] also tackled the profit maximization models of data brokers, which consider both the willingness-to-buy of personal data consumers and the willingness-to-sell of personal data providers. These studies showed the feasibility of personal data trading markets that consider the consent and active participation of personal data providers.

Many previous studies assumed that data brokers provide newly analyzed datasets to data consumers. However, data consumers now want to buy raw personal data to apply their own analysis techniques to provide new services for finding new potential customers. Now, it is necessary to check the behavioral model of a data consumer to maximize its profit by providing new services with the collected dataset from data providers, which is affected by data providers' behavior related to explicit consent.

Therefore, this paper proposes a profit maximization model for a data consumer when it buys personal data from data providers (by obtaining consent) through data brokers and provides their new services to data providers (e.g., target advertisement, etc.). Particularly, this paper jointly considers the behavior of data providers (WTS) and service users (WTP) under a limited budget for the data consumer, in which the budget affects both the total size of the collected dataset from data providers and the service quality produced by the collected dataset for service users. The detailed contributions of this paper are as follows.

- This paper proposes a personal data trading model considering payments of direct incentives to the data providers, in which a data consumer collects personal data from data providers with explicit consent about providing direct incentives through data brokers and creates a new service for targeting the same data providers.
- This paper proposes new revenue and cost models for a data consumer by considering the behavioral models (i.e., willingness-to-pay of service users and willingness-to-sell of data providers) adopted from the authors' previous works [25,26], which are inspired by the real-world surveys and observations [6,28].
- According to the revenue and cost models, this paper proposes a profit maximization problem. By applying convex optimization techniques, this paper transforms the problem to be more practical. Consequently, this paper finally proposes a constrained profit maximization problem with the limited budget of the data consumer under the practical boundary of cost allocation, which is a constrained nonlinear optimization problem.
- With parameters inspired by real-world survey [29] on data providers, which provided the results with about 1000 respondents regarding willingness-to-share their personal information in exchange for money (i.e., willingness-to-sell personal data), this paper shows various numerical results to check the feasibility of the proposed models. Moreover, this paper identifies several discussion points regarding the proposed model and the analytical results.

The rest of this paper is organized as follows; Section 2 shows relevant studies on data trading markets with/without incentives for data providers. In Section 3, this paper describes a considered personal data trading model with data providers' incentives, including major stakeholders (i.e., data providers, data brokers, a single data consumer, and service users). In addition, the mathematical models are formulated to represent the characteristics of the stakeholders (including WTS, WTP, etc.). Using the models, Section 4 proposes a profit maximization problem of the data consumer considering both revenues with WTP of service users and costs with WTS of data providers. By analyzing the proposed problem, this paper derives a constrained profit maximization problem with a limited budget, which has a practical boundary for the data consumer. To show the feasibility of the proposed problem, in Section 5, numerical results with both theoretical and practical parameters are analyzed to check how the profit of the data consumer is decided. Finally, this paper is concluded in Section 6.

## 2. Related Work

Many surveys have discovered and analyzed data pricing schemes and data markets [1–3]. These surveys identified various characteristics of data markets (e.g., business models, market structures, stakeholders, lifecycle, etc.).

The concept of WTP or WTB of data consumers has also been mainly considered in various data markets. Particularly, in the field of engineering, Internet of Things (IoT) based data markets have been mainly considered because IoT devices (e.g., sensors, wearable devices) act as data sources for data providers. Niyato et al. [6] proposed a simple Internet of Things (IoT) data market model, which considered the WTB of data consumers depending on data quality. It also proposed mathematical models in which the data price is related to the service quality, which in turn depends on the data size. This work provided a basic mathematical model for the willingness-to-buy and service quality model, but it only considered a single data broker for the data market. Yoshihiro and Hosio [7] proposed a simulation-based IoT data pricing model that considers the WTB of data buyers. A data broker, which intermediates between seller and buyer, decides an optimal data price on the balance between the seller's competition and the buyer's demand. It investigated a sensor data market model with a fixed selling price decided by data sellers which cannot be changed when the buyer actually chooses the sellers. Moreover, with the concept of WTP (or WTB), competitive data markets also have been actively studied. Jang et al. [8] modeled an IoT data market with multiple independent data sources. In this model, a

data broker has a limited amount of budget to buy data from all data sources with the non-cooperative data trading model. Seo et al. [9] proposed an ICN-based Data market model with a non-cooperative competition game for profit maximization of a data broker. It proposed a concept of nominal WTP that allows data consumers to discover required data with the required quality, and it also proposed an expected service quality function with respect to data size. These studies provided non-cooperative competition games only for data brokers; however, they did not tackle the behavioral characteristics of data providers.

However, the aforementioned studies only considered WTB have mainly considered IoT data markets, and they did not consider the personal data (or privacy) market that should be considered in the personal data trading model. Parra-Arnau [23] investigated the trade-off between privacy and money of data providers and proposed an optimization model for profile-disclosure risks and economic rewards. Su et al. [24] proposed an incentive-based crowd-sourcing scheme for collecting various data in cyber-physical-social systems with an auction-based price bidding scheme for data providers. The above studies considered the willingness-to-sell model of data providers, but they only considered the relationship between data providers and data brokers, not data consumers.

Moreover, Oh et al. [25,26] (the authors' previous studies) proposed profit maximization models of data brokers by considering both the willingness-to-buy of personal data consumers and the willingness-to-sell of personal data providers in normal and competitive personal data markets, respectively. Moyopo and Qu [27] also proposed an IoT data trading model that includes the WTS of data providers, the WTB of data consumers, and a data quality model that incorporates data currency. These studies considered the behavioral models for both data providers and data consumers; however, they have mainly focused on the problem of data brokers considering data providers and data consumers.

Unlike the previous studies, this paper proposes a profit maximization model for the data consumer and its business model that utilizes the data brokers to obtain explicit consent for providing direct incentives and personal data providers as sources of personal data and provides new services to the potential service users for earning revenues.

## 3. System Model

This section describes a consent-based personal data trading model for data consumers. Usually, data brokers have collected personal data according to the opt-in-based agreement from data providers. However, to utilize personal data with a new purpose, it is necessary to obtain new explicit consent from data providers. Therefore, this paper considers a consent-based personal data trading model, which is illustrated in Figure 1.

This paper considers a data trading market consisting of four groups that behave for their own benefit.

- Data providers: a group of candidate individuals who may provide their own personal data according to their informed consent of using personal data while receiving direct incentives as compensation;
- Data brokers: a group of data brokers who participate data trading market to intermediate between data providers and the data consumer;
- Data consumer: a data consumer who processes personal data and creates new services for service users. The data consumer wants to maximize its profit by providing good quality services to service users while minimizing budget consumption (note that it is a main target for profit maximization analysis in this paper);
- Service users: a group of candidate individuals who may use a service that is provided by the data consumer.

In this model, a data consumer buys personal data from data providers via data brokers. A data broker intermediates both data providers and the data consumer to aggregate personal data and have personal data trade settled and paid. When the data consumer tries to buy personal data, the data consumer offers the price of the required personal data to each data provider to obtain new consent. When data providers receive an offer from the data consumer, data providers decide whether they accept the offer or

not by weighing the value of the offer and the privacy sensitivity of the personal data. After collecting personal data from data brokers, the data consumer analyzes the collected dataset and creates a new service. With the created service, it sells to potential service users who decide to use the service if it satisfies the price condition.

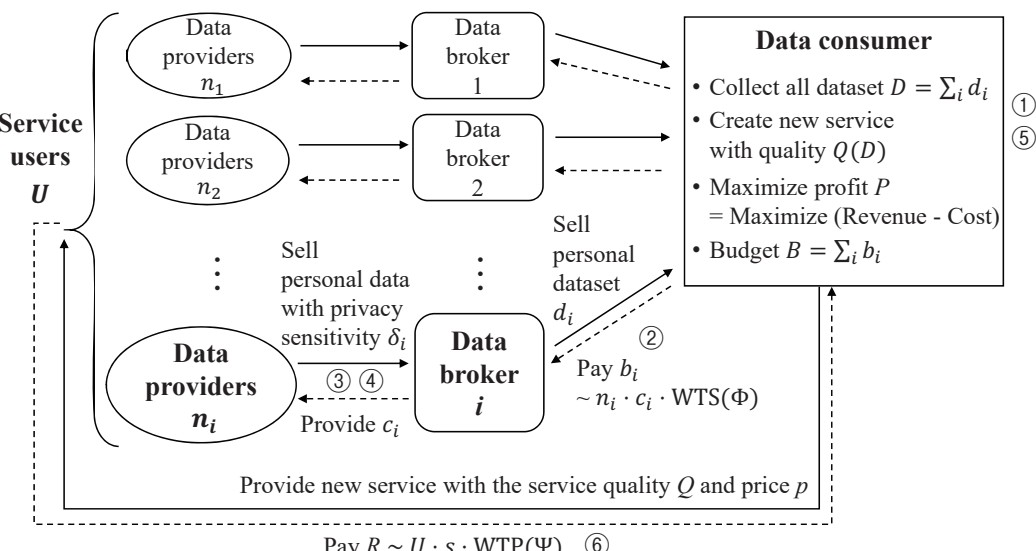

① With budget $B$, the data consumer estimates the achievable maximum profits.
② Pay $b_i$ for a data broker $i$ to obtain dataset $d_i$.
③ The data providers $n_i$ sell their personal data and sell to the data broker.
④ The data broker distributes direct incentive $c_i$ to the data providers.
⑤ The data consumer aggregates all dataset $D$ and create a new service with quality $Q(D)$.
⑥ The data consumer provides new service with the service quality $Q$ and price $p$.

**Figure 1.** An illustration of the proposed consent-based personal data trading model.

Under this environment, a major objective of the data consumer is to maximize its profit with the consideration of both revenues for providing good quality service and costs for buying personal data. Therefore, this paper proposes the profit maximization problem for the data consumer, which jointly considers the behavior of data providers and service users. Particularly, this section formulates base models of stakeholders (i.e., data providers, data consumers, and service users) before analyzing profit maximization problems.

### *3.1. Cost Model of Data Consumer*

According to the considered system model, the data consumer buys personal data from data brokers, and then it creates a new service to sell. Therefore, this section first introduces a cost model for buying personal data from data providers via data brokers.

### 3.1.1. Willingness-to-Sell of Data Providers

The data consumer needs to aggregate datasets from data providers through data brokers to perform data analysis. To collect personal data, the data consumer needs to obtain informed consent from data providers for the purpose of personal data use by offering proper prices to each data provider. Therefore, the willingness-to-sell of data providers should be considered to model the expected cost of the data consumer because the data consumer does not know the exact number of data providers who agree to the data use proposal of the data consumer.

Since the behavior of individual data providers is hard to model [25], similar to the previous study, this paper models a macro-level willingness-to-sell model of data providers by considering the offered price and the privacy sensitivity of the required personal data. In general, if the offered price is high, then the portion of data providers increases. On the other hand, if the required personal data are highly privacy sensitivity, then the portion

of data providers with the same offered price decreases. In other words, the portion of data providers slowly increases if the required personal data have high privacy-sensitive information; otherwise, the portion of them rapidly increases. Based on these principles, similar to the previous studies [25,26], the WTS of data providers with the offered price and the privacy sensitivity of the required personal data is defined as follows.

**Definition 1** (Willingness-to-sell of data providers)**.** *The WTS function $\Phi$ of the data providers in a data broker is defined by the cumulative distribution function as the portion of the data providers based on the offered price $c_i$ from the data consumer and the privacy sensitivity of the required data $\delta_i$, and given by*

$$\Phi(c_i, \delta_i) = 1 - e^{-c_i(1-\delta_i)}, \tag{1}$$

*where $c_i > 0$ and $0 < \delta_i < 1$.*

The trends of the proposed WTS model are shown in Figure 2a. The WTS increases when the offered price $c_i$ increases. Moreover, the trends of WTS have different slopes with respect to the privacy sensitivity factor $\delta$. If $\delta$ becomes large, it means the required personal data is more privacy-sensitive (i.e., people be more skeptical to share their personal data); in other words, data providers hesitate to sell. On the other hand, if $\delta$ becomes small, it means the required personal data is less privacy-sensitive; in other words, data providers are more decisive in selling.

A privacy sensitivity parameter $\delta$ represents multiple meanings. One is the behavior related to data types. According to surveys [28], people have different privacy awareness for each data type (e.g., people are more hesitant to share their medical records than their name or age). Moreover, when people are asked to share a combination of multiple data types (or when the number of required data types increases), they are more hesitant (e.g., people are more hesitant to share age and name than their age solely).

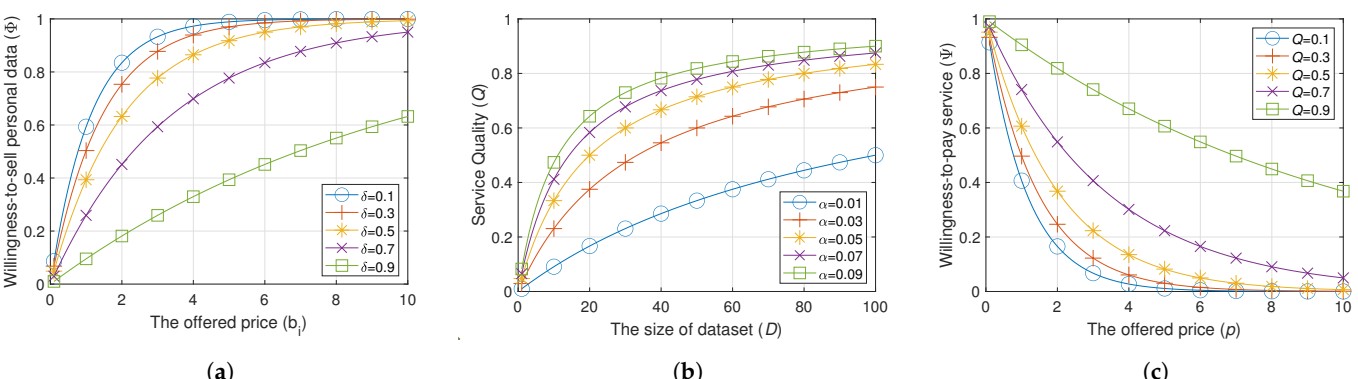

(**a**)  (**b**)  (**c**)

**Figure 2.** The trends of the proposed mathematical models inspired by the real-world surveys and observations [6,25,28]. (**a**) Willingness-to-sell of data providers. (**b**) Service quality of data consumer. (**c**) Willingness-to-pay of service users.

3.1.2. The Size of Collected Dataset

Since the data consumer is able to estimate the expected number of actual data providers using the WTS model, the data consumer now can forecast the costs of buying personal data from a data broker. According to the WTS model, the expected size of the personal dataset (i.e., the number of actual data providers) from a data broker $d_i$ can be represented as follows.

$$d_i = n_i \Phi(c_i, \delta_i), \tag{2}$$

where $n_i$ means the entire number of potential data providers associated with a data broker and $\Phi(c_i, \delta_i)$ means the WTS (i.e., the portion of the consented data providers). $d_i$ means an obtainable size of the dataset from the data broker $i$. Note that, for personal data utilization, it is important to collect information about different individuals as much as

possible. Therefore, in this paper, the number of data providers is proposed to measure the relative data size.

Then, the expected size of the personal dataset from all data brokers ($D$) can be represented as a summation of the dataset from each data broker.

$$
\begin{aligned}
D = \sum_i d_i &= \sum_i n_i \Phi_i(c_i, \delta_i), \\
&= \sum_i n_i (1 - e^{-c_i(1-\delta_i)}).
\end{aligned} \tag{3}
$$

Note that, in this paper, the possible overlapped information of data providers from different data brokers is not considered for the simplicity of the problem.

### 3.1.3. The Expected Costs of the Data Consumer

Since the cost $c_i$ is paid to each data provider, the expected required budget $b_i$ of the data consumer to obtain the dataset $d_i$ from one data broker is represented as follows.

$$
\begin{aligned}
b_i = c_i d_i &= c_i n_i \Phi(c_i, \delta_i), \\
&= c_i n_i (1 - e^{-c_i(1-\delta_i)}).
\end{aligned} \tag{4}
$$

Therefore, the expected cost of the data consumer to obtain the dataset $D$ from the entire data providers is represented as follows.

$$
C(\bar{c}) = \sum_i b_i = \sum_i c_i n_i (1 - e^{-c_i(1-\delta_i)}), \tag{5}
$$

where $\bar{c} = \{c_1, \cdots, c_i\}$ means a set of cost allocation for each data provider, $\bar{n} = \{n_1, \cdots, n_i\}$ means a set of the number of data providers, and $\bar{\delta} = \{\delta_1, \cdots, \delta_i\}$ means a set of privacy sensitivity factors of data providers in data brokers.

### 3.2. Revenue of Data Consumer

After buying personal data from data providers, the data consumer creates a new service (e.g., target advertisement, new goods recommendations, etc.) by analyzing the collected personal data. Then, it sells the new service to potential service users to earn revenue. Therefore, this section introduces a revenue model of the data consumer that considers the expected service quality obtained from the collected dataset and the expected service users.

### 3.2.1. A Service Quality of the Data Consumer

Since the data consumer needs to provide good quality services to the service consumers to maximize its revenue, it is necessary to consider the expected service quality model with respect to aggregate datasets from data providers. There are many ways to define the quality of services produced by personal data analysis. Today, one of the featured applications is artificial intelligence (AI) and machine learning (ML). Particularly, an objective of AI/ML is increasing accuracy for a certain problem (e.g., classification, clustering, retrieval, etc.). Moreover, according to surveys [30], many service consumers expect to have a better customer experience with more accurate recommendations, advertisements, etc. in exchange for their personal data, which is related to service accuracy. Therefore, similar to the previous studies [6,9,26] on data trading, this paper also considers an accuracy-based service quality model for the data consumer.

Similar to the previous studies, for modeling the service quality function, two principles are mainly considered. One is that service quality increases when the size of input data increases. The other is the law of diminishing marginal utility (i.e., the marginal service quality decreases when the size of the dataset increases) [31]. According to the principles,

the expected service quality function $Q$ of the data consumer with the aggregated dataset $D$ is defined as follows.

**Definition 2** (The expected service quality with aggregated dataset).

$$Q(D) = 1 - \frac{1}{1 + \alpha D},$$

(6)

*where $\alpha \in (0, 1]$ and $D > 0$.*

In Equation (6), $\alpha$ represents a curve-fitting parameter, and $D$ represents the size of dataset. $Q(\cdot)$ is a non-decreasing function with decreasing marginal accuracy, and the higher $Q$ value indicates better service quality. Note that the form of the proposed service quality model is also used in several previous studies [6,9,26], which mathematically modeled several accuracy curves of the big data processing and machine learning results by analyzing the real-world datasets. In this paper, the data consumer has zero knowledge about the potential data providers and/or service users; therefore, the service quality value becomes zero when there is no data (i.e., $Q(0) = 0$).

The parameter $\alpha$ means the quality of data that affects the accuracy of data processing methods (i.e., if $\alpha$ becomes high, the accuracy rapidly increases). For example, if a dataset consists of many data types, it is more helpful to increase machine learning accuracy rather than that of a few data types. Figure 2b shows a trend of the proposed service quality model $Q$ with respect to the parameter $\alpha$ and the size of dataset $D$.

3.2.2. Willingness-to-Pay of the Service Consumers

To estimate a profit model of a data consumer, the expected revenue should be considered. From the data consumer's perspective, one of the possible approaches is estimating the expected number of service users when the data consumer provides a new service to them. However, the behavior of each service consumer is hard to directly formulate because the decision criteria of each service consumer vary. Therefore, many relevant studies have considered a macro-level model to formulate the behavior of service consumers. The willingness-to-pay model is one of the well-known models in data trading studies [6,9,26].

Similar to the related studies, this paper considers a willingness-to-pay model that considers the price and quality of the provided service. The principle of the proposed willingness-to-pay model follows two characteristics. One is that WTP decreases when service price increases. The other is that WTP gradually decreases when the quality of service is high; in other words, WTP rapidly decreases when the quality of service is low, representing the relationship between WTP and service quality.

**Definition 3** (Willingness-to-pay of service consumers). *The WTP function $\Psi$ of the service consumers is defined by the cumulative distribution function as the portion of the paid services consumers based on the offered price $s$ and the service quality $Q$, and given by*

$$\Psi(s, Q) = e^{-s(1-Q)},$$

(7)

*where $s > 0$ and $0 \leq Q < 1$.*

In Equation (7), $s$ represents an offered service fee to service users, and $Q$ represents the service quality provided by the data consumer. Figure 2c shows the trends of the proposed WTP model. The WTP decreases when an offered price $s$ increases; on the other hand, if the service quality is relatively low (e.g., $Q = 0.1$), the WTP rapidly decreases while the offered price increases. Meanwhile, if the service quality is relatively high (e.g., $Q = 0.9$), the WTP gradually decreases while the offered price increases.

### 3.2.3. The Expected Revenue of Data Consumer

By combining Equations (6) and (7), the expected revenue of the data consumer $R(s)$ with the offered service price $s$, which utilizes datasets and provides services to the potential service users with a certain service fee, is represented as follows:

$$R(s) = Us\Psi(s, Q),$$
$$= Use^{-s(1-Q)}, \tag{8}$$

where $U$ means the potential service users, $s$ and $Q$ mean the price and the service quality of the new service provided by the data consumer, respectively, and $\Psi$ means the willingness-to-pay of service users. $U\Psi(s, Q)$ means the number of paid service users; therefore, the total revenue becomes service fee $s$ multiplied by the number of paid service users.

## 4. Proposed Personal Data Trading Model for Data Consumer

In the previous section, this paper formulates mathematical models for data providers, the data consumer, and service users to see the characteristics related to the profit of the data consumer. This section proposes a profit maximization problem considering the behavior models introduced above.

### 4.1. Profit Maximization Model for Data Consumer

From the data consumer's point of view, it needs to achieve the maximum profit by maximizing revenues and minimizing costs. Therefore, it is necessary to consider the profit maximization model for the data consumer with the consideration of the revenue and cost models described in the previous section. Based on the revenue and cost models of the data consumer, the profit function $P$ of the data consumer is defined as follows:

$$P(s, \bar{c}) = R(s) - C(\bar{c}), \tag{9}$$

where $R(\cdot)$ means the expected revenue in Equation (8) and $C(\cdot)$ means the expected costs in Equation (5).

Profit Maximization Problem

This paper assumes that marginal costs (e.g., additional costs for data management, etc.) are negligible. For simplicity, this paper also assumes that the fees to intermediate transactions between data consumers and data providers are included in a cost ($\bar{c}$).

Then, the profit maximization problem is formulated as follows. **Problem 1**: *Profit maximization problem*

$$\max_{s,\bar{c}} P(s, \bar{c}) \tag{10}$$

$$s.t. \quad P > 0, \tag{11}$$

$$n_i > 0 \quad \forall n_i \in \bar{n}, \quad U > 0, \tag{12}$$

$$0 < \alpha \le 1, \quad 0 < \delta_i < 1 \quad \forall \delta_i \in \bar{\delta}, \tag{13}$$

$$\forall c_i \in \bar{c} > 0, \tag{14}$$

$$s > 0. \tag{15}$$

The proposed **Problem 1** can be formed under the conditions (11) to (15). The condition (11) means that the data consumer only considers the problem with positive profit values. The condition (12) is about the number of data providers and service users, which should be larger than 0. The next condition (13) indicates the range of parameters for privacy sensitivity and service quality. The costs for buying personal data and the price for a new service are indicated in conditions (14) and (15), respectively.

In this section, the proposed profit maximization problem for the data consumer is solved. First, the concaveness of the revenue function $R$ is checked to find the optimal

service price $s^*$ to maximize revenue. To check the concaveness of the revenue function $R$ with $s$, it is checked by considering whether the second-order derivative of $R$ with $s$ is less than zero or not.

$$\frac{\partial}{\partial s}R = Ue^{s(Q-1)} + Use^{s(Q-1)}(Q-1),$$

$$\frac{\partial}{\partial^2 s}R = 2Ue^{s(Q-1)} + Use^{s(Q-1)}(Q-1)^2,$$

$$\Rightarrow -Ue^{s(Q-1)}(1-Q)(1+Q) < 0.$$

Note that $Ue^{s(Q-1)}(1-Q)(1+Q)$ always be positive according to the conditions in **Problem 1**.

Since the revenue function $R(\cdot)$ is a concave function, the maximum service price point $s^*$ is obtained by checking the first-order derivative with $s$ becomes zero.

$$\frac{\partial}{\partial s}R = Ue^{s(Q-1)}(1 + s(Q-1)) = 0,$$

$$\Rightarrow 1 - s^*(1-Q) = 0,$$

$$\therefore s^* = \frac{1}{1-Q}.$$

By applying the $s^* = \frac{1}{1-Q}$ to the revenue function $R(\cdot)$, then the transformed revenue function $R^*(\cdot)$ is obtained as follows.

$$R^*(s^*) = Us^* e^{-s^*(1-Q)},$$

$$= U\frac{1}{(1-Q)}e^{-\frac{1}{(1-Q)}(1-Q)},$$

$$= Ue^{-1}(1 + \alpha D),$$

$$= Ue^{-1}(1 + \alpha \sum_i n_i \Phi(c_i, \delta_i)).$$

Note that the size of dataset $D$ (in Equation (3)) is applied. Then, the transformed revenue function $R^*(\cdot)$ is defined as follows.

$$R^*(\bar{c}) = Ue^{-1}(1 + \alpha \sum_i n_i \Phi(c_i, \delta_i)). \tag{16}$$

By applying the $s^*$, the transformed revenue function also becomes the function of budget allocation $\bar{c}$.

Since $R^*$ is also the function of $\bar{c}$ based on the equations, the profit function $P(\cdot)$ can be represented as $P^*(\bar{c})$ as follows,

$$P^*(\bar{c}) = R^*(\bar{c}) - C(\bar{c}),$$

$$= Ue^{-1}\left(1 + \alpha \sum_i n_i \Phi(c_i, \delta_i)\right) - \sum_i c_i n_i \Phi(c_i, \delta_i). \tag{17}$$

Then, the original profit maximization problem can be transformed as follows based on the new profit function $P^*(\cdot)$. **Problem 2**: *A transformed profit maximization problem*

$$\max_{\bar{c}} P^*(\bar{c}) \tag{18}$$

$$s.t. \quad \text{conditions } (11) - (14).$$

Note that the service price condition (15) is expelled for the proposed **Problem 2**. Similar to the original problem, if the transformed profit function $P^*(\cdot)$ is concave, it means

that the existence of the unique combination of $\bar{c}$ represents the maximum profit value. Therefore, the concaveness of the new profit function is also checked.

To check the concaveness of the transformed profit function $P^*$ with $c_i$, the profit function $P^*(\cdot)$ is rearranged for $c_i$ and the rest of elements in $\bar{c}$ as follows.

$$
\begin{aligned}
P^*(\bar{c}) = {} & \alpha U e^{-1} n_i (1 - e^{-c_i(1-\delta_i)}) - n_i c_i (1 - e^{-c_i(1-\delta_i)}) \\
& + \alpha U e^{-1} \sum_{k \neq i} n_k \Phi(c_k, \delta_k) - \sum_{k \neq i} n_k c_k \Phi(c_k, \delta_k) + U e^{-1}.
\end{aligned} \tag{19}
$$

Note that checking the concaveness of the multivariable function can be done by checking the concaveness of the function of one variable [32]. Therefore, the first and the second order derivative of the $P^*(\cdot)$ (in Equation (19)) with $c_i$ are checked as follows. Note that $(e^{kx})' = k e^{kx}$ and $(x e^{kx})' = e^{kx} + k x e^{kx}$.

$$
\begin{aligned}
& \frac{\partial}{\partial c_i} P^*(\bar{c}) \\
& \quad = \alpha(1-\delta_i) U e^{-1} e^{-c_i(1-\delta_i)} - n_i \\
& \qquad + n_i e^{-c_i(1-\delta_i)} - n_i(1-\delta_i) c_i e^{-c_i(1-\delta_i)}, \\
& \quad = \{\alpha(1-\delta_i) U e^{-1} + n_i\} e^{-c_i(1-\delta_i)} \\
& \qquad - n_i(1-\delta_i) c_i e^{-c_i(1-\delta_i)} - n_i, \\
& \frac{\partial}{\partial^2 c_i} P^*(\bar{c}) \\
& \quad = -\{\alpha(1-\delta_i) U e^{-1} + n_i\} (1-\delta_i) e^{-c_i(1-\delta_i)} \\
& \qquad - n_i(1-\delta_i) e^{-c_i(1-\delta_i)} \\
& \qquad + n_i(1-\delta_i)(1-\delta_i) c_i e^{-c_i(1-\delta_i)}, \\
& \quad = (1-\delta_i) e^{-c_i(1-\delta_i)} \{(1-\delta_i)(-\alpha U e^{-1} + n_i c_i) - 2n_i\}.
\end{aligned}
$$

(with equation numbers (20) and (21))

For satisfying the concaveness condition, the second-order derivative (in Equation (21)) becomes less than zero. Therefore, the following should be satisfied.

$$
\underbrace{(1-\delta_i)}_{>0} \underbrace{(e^{-c_i(1-\delta_i)})}_{>0} \{(1-\delta_i)(-\alpha U e^{-1} + n_i c_i) - 2n_i\} < 0,
$$

$$
\Rightarrow \{(1-\delta_i)(-\alpha U e^{-1} + n_i c_i) - 2n_i\} < 0,
$$

$$
\Rightarrow c_i < \frac{\alpha(1-\delta_i) U e^{-1} + 2n_i}{n_i(1-\delta_i)},
$$

$$
\Rightarrow c_i < \frac{\alpha U}{n_i} + \frac{2}{(1-\delta_i)}. \tag{22}
$$

According to this analysis, the Equation (22) is obtained. This boundary condition gives an insight into allocating a cost and budget for buying personal data. The first term $\frac{\alpha U}{n_i}$ means the portion of the number of paid service users and the number of data providers necessarily large enough to allocate enough cost for each data provider; in other words, it can be used as the boundary of possible profit with achievable service quality $\alpha$. On the other hand, the second term $\frac{2}{(1-\delta_i)}$ means the upper limit of the payable price if there is no service user (i.e., $U = 0$); in other words, it indicates the maximum unit cost to buy personal data from a data broker.

This boundary condition (Equation (22)) gives the feasible range of $c_i$ to the data consumer. Since the proposed WTS model is an asymptotic function, if there is no boundary for $c_i$, it may affect solving the proposed profit maximization problem as bias. The effects of the obtained boundary condition are also analyzed in the next section.

Unfortunately, it is not possible to find a unique solution that makes the first-order derivative of the transformed profit function (in Equation (20)). Therefore, this paper proposes another transformed problem; that is, a constrained profit maximization problem with a budget constraint by limiting the total amount of budget of the data consumer.

### 4.2. Constrained Profit Maximization Problem

The basic assumption of the proposed **Problem 1** and **Problem 2** is that the data consumer does not consider the total amount of available budget to buy personal data from data providers in the market. However, in reality, the data consumer usually has a budget constraint. Therefore, this constraint budget becomes the maximum available cost for the data consumer. In addition, the cost allocation for data providers also should be feasible (explained in Equation (22)). By considering such constraints, the transformed profit maximization problem (**Problem 2**) is now considered a constrained profit maximization problem (**Problem 3**).

**Problem 3**: *Constrained profit maximization problem*

$$\max_{\bar{c}} P^*(\bar{c}) \tag{23}$$

$$s.t. \quad \text{conditions } (11) - (13),$$

$$0 < c_i < \frac{\alpha U}{n_i} + \frac{2}{(1 - \delta_i)}, \tag{24}$$

$$B \geq \sum_i n_i c_i (1 - e^{-c_i(1 - \delta_i)}). \tag{25}$$

In **Problem 3**, $B$ indicates the total amount of budget of the data consumer. The condition (24) means the cost boundary for buying personal data from one data provider, and the condition (25) means the total budget boundary for buying personal data from all data brokers.

Note that the cost and budget are now becoming a constraint of the problem (in conditions (24) and (25)). By bounding cost and budget, now **Problem 3** becomes a constrained nonlinear optimization problem [33]. Therefore, any constrained nonlinear optimization algorithms can be used to solve the problem under the conditions. Similar to the authors' previous study [26], this paper utilizes the sequential least squares programming (SLSQP) method supported by SciPy (a widely used library for scientific Python) [34] for checking numerical results, which is performed in the next section. For applying the algorithm, this paper solves "$\min_{\bar{c}} -P^*(\bar{c})$" because the SLSQP algorithm only supports the minimization approach.

## 5. Numerical Results

This section analyzes the proposed personal data trading model for the data consumer. This paper observes the various numerical trends of the proposed profit maximization problems (**Problem 2** and **Problem 3**). Particularly, the **Problem 2** is related to a theoretical analysis without costs and budget constraints, and **Problem 3** is related to a practical analysis with real-world parameters. Note that, unfortunately, there are only few studies that have tackled direct incentives to data providers, and they considered different environments; therefore, it is not able to provide a direct comparison between the proposed model and other models.

### 5.1. Theoretical Analysis

For setting the privacy sensitivity parameter $\delta$, this paper assumes that 50% of the data providers participate in the market with the average price (i.e., $\Phi(\cdot) = 0.5$). By re-arranging variables in Equation (1), it is possible to obtain each $\delta_i$ as follows: $\delta_i = \frac{ln(0.5)}{c_i} + 1$. For reader's information, if the average costs ($c_i$) are $1.0 and $90.0, then the values of $\delta$ become 0.3069 and 0.9923, respectively. Therefore, for the theoretical analysis, this paper uses the values of $\delta$ between 0.3 to 0.99.

The effects of other parameters such as the number of service users $U$ and data providers $n_i$ with a data broker and the curve fitting parameter for service quality function $\alpha$ are also analyzed in the following section. Particularly, the number of service users $U$ is set to the total number of data providers in the market (i.e., $U = \sum_i n_i$) because the data consumer buys personal data from data providers and then tries to provide a new service to the same group.

5.1.1. Data Trading with a Single Data Broker

This section observes the trends of the profit values (in Equation (17) for **Problem 2**) with a single data broker. Particularly, by observing the phenomenons, it mainly shows the effects of the parameters (i.e., privacy sensitivity parameter $\delta$ and service quality parameter $\alpha$).

For this experiment, this paper sets the variables for the number of service users $U$ and data providers $n_1$ with the single data broker as 500 (i.e., $U = n_1 = 500$), which affects only the size of the profit values. Figure 3 shows the trends of the profit for the data consumer with respect to various offered prices $c_i$ with different privacy sensitivity factor $\delta$ (Figure 3a) and service quality factor $\alpha$ (Figure 3b).

Figure 3a shows the trends of the profit for the data consumer with different privacy sensitivity factors $\delta$. With the fixed service quality factor ($\alpha = 0.5$), the maximum profits are shown as follows:

$$\begin{cases} \bar{\delta} = \{0.3\} : 4.25 \times 10^4 \text{ at } \bar{c} = (5.9), \\ \bar{\delta} = \{0.7\} : 3.92 \times 10^4 \text{ at } \bar{c} = (10.8), \\ \bar{\delta} = \{0.9\} : 3.13 \times 10^4 \text{ at } \bar{c} = (20.9). \end{cases}$$

The profit value of the data consumer becomes larger when the data broker handles less private personal data (i.e., smaller $\delta$) because the number of data providers who actually participate in the market is larger, which is directly related to the size of the dataset.

Since the same value of $\alpha$ means that the data consumer creates the same quality of service regardless of the costs of buying personal data, the proposed service quality function is directly affected by the size of the collected dataset. In other words, with a larger dataset, the data consumer can generate a service with better quality, and it makes more service consumers pay for the newly created service. Therefore, the profit of buying less privacy-sensitive personal data becomes larger than that of buying more privacy-sensitive ones.

On the other hand, Figure 3b shows the trends of the profit for the data consumer with various values of the service quality parameter $\alpha$. With the fixed privacy sensitivity parameter ($\delta = 0.9$), the maximum profits are shown as follows:

$$\begin{cases} \alpha = 0.1 : 3.06 \times 10^3 \text{ at } \bar{c} = (7.4), \\ \alpha = 0.5 : 3.13 \times 10^4 \text{ at } \bar{c} = (20.9), \\ \alpha = 0.9 : 6.48 \times 10^4 \text{ at } \bar{c} = (27.0). \end{cases}$$

The profit value of the data consumer becomes larger when the service quality parameter becomes larger. With the service quality parameter $\alpha = 0.1$, the data consumer makes $3.06 \times 10^3$ profit with a cost of 7.4. On the other hand, with $\alpha = 0.9$, the data consumer makes $6.48 \times 10^4$ with a cost of 27.0.

Since the data consumer can generate better service with a higher value of service quality parameter, the data consumer can make more profit even though it consumes more costs for buying personal data with the same privacy sensitivity parameter.

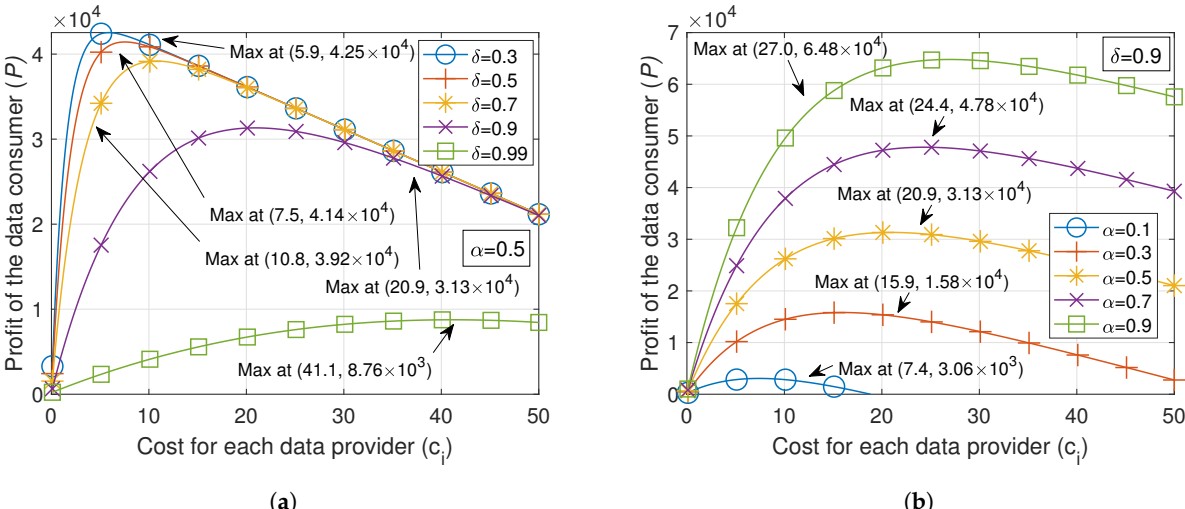

**(a)**

**(b)**

**Figure 3.** The trends of the profit for data consumers with the single data broker. (**a**) Privacy sensitivity $\delta$. (**b**) Service quality parameter $\alpha$.

5.1.2. Data Trading with Two Data Brokers

Next, this section checks the trends of the profit values (in Equation (17) for **Problem 2**) for the data consumer with two data brokers. Particularly, the profits of the data consumer with different privacy sensitivity parameters and the different number of data providers are analyzed.

Firstly, similar to the above with the single data broker, the profits of the data consumer with different privacy sensitivity parameters $\delta$ are observed in Figure 4. Note that the service quality parameter $\alpha$ is set to 0.9 (i.e., $\alpha = 0.9$). With the same number of data providers ($n_1 = 500$ and $n_2 = 500$) and service users ($U = 1000$), the profit values of the data consumer are shown as follows:

$$\begin{cases} \bar{\delta} = \{0.9, 0.7\} : 3.00 \times 10^5 \text{ at } \bar{c} = (34.2, 15.2), \\ \bar{\delta} = \{0.9, 0.9\} : 2.88 \times 10^5 \text{ at } \bar{c} = (34.2, 34.2), \\ \bar{\delta} = \{0.9, 0.99\} : 1.99 \times 10^5 \text{ at } \bar{c} = (34.2, 50.0). \end{cases}$$

The trends of profit values for the data consumer with two data brokers are similar to the combined profit values with two different single data brokers. The profit value of data consumer decreases and costs for buying personal data increases when a data broker has higher privacy sensitivity.

The case with a single data broker and the case with two data brokers show similar trends of profit values for the data consumer; however, the data consumer can make more profit when it is able to buy personal data from two data brokers under the same condition (i.e., the same parameters $\delta$ and $\alpha$).

For example, with the same privacy sensitivity parameter ($\delta_i = 0.9$) and the same service quality parameter ($\alpha = 0.9$), the data consumer makes $6.48 \times 10^4$ profit with $\bar{c} = (27.0)$ for the single data broker case, but it makes $2.88 \times 10^5$ profit with $\bar{c} = (34.2, 34.2)$ for the two data broker cases.

On the other hand, the trends of profit values for the data consumer with the different number of data providers and service users are analyzed in Figure 5. Note that the service quality parameter $\alpha$ is set to 0.5 (i.e., $\alpha = 0.5$). With the same privacy sensitivity of data brokers ($\delta_1 = \delta_2 = 0.9$), the profit values of the data consumer are shown as follows:

$$\begin{cases} \bar{n} = \{250, 500\}, U = 750 : 7.80 \times 10^4 \text{ at } \bar{c} = (25.1, 25.1), \\ \bar{n} = \{500, 500\}, U = 1000 : 1.47 \times 10^5 \text{ at } \bar{c} = (28.1, 28.1), \\ \bar{n} = \{750, 500\}, U = 1250 : 2.38 \times 10^5 \text{ at } \bar{c} = (30.4, 30.4). \end{cases}$$

The number of data providers and service users directly affects the profit value of the data consumer. When the number of data providers and service users increases, the profit value also increases with the increased costs for buying personal data from each data broker.

The case with a single data broker and the case with two data brokers show similar trends of profit values for the data consumer; however, the data consumer can make more profit when it is able to buy personal data from two data brokers under the same condition (i.e., the same parameters $\delta$ and $\alpha$, and the same number of data providers in each data broker $\bar{n}$ with the same number of service users $U$).

For example, with the same privacy sensitivity parameter ($\delta_i = 0.9$), the same service quality parameter ($\alpha = 0.5$), and the same number of data providers ($n_i = 500$) with the same number of service users ($U = 1000$), the data consumer makes $3.13 \times 10^4$ profit with $\bar{c} = (20.9)$ for the single data broker case, but it makes $1.47 \times 10^5$ profit with $\bar{c} = (28.1, 28.1)$ for the two data broker cases.

The results of theoretical analysis in this section are good for observing and checking the trends of profit values for the data consumer with respect to various variables. However, there are some limitations in the theoretical analysis; for example, in Figure 3a, the WTS value becomes 0.98 at $c_1 = 5.9$ with $\delta = 0.3$ that resulted in the maximum profit of $4.25 \times 10^4$, which means the about 98% of the total data providers (in this case, about 490 out of 500) accepts the offered price to buy personal data.

However, in the real world, some portions of the entire data providers (i.e., about 10–30% of data providers [26]) do not participate in the personal data trading market for various reasons. Therefore, in the next section, this paper analyzes the proposed constrained profit maximization problem (**Problem 3**) with budget and cost allocation constraints for the data consumer with the real-world survey-based privacy sensitivity parameter values.

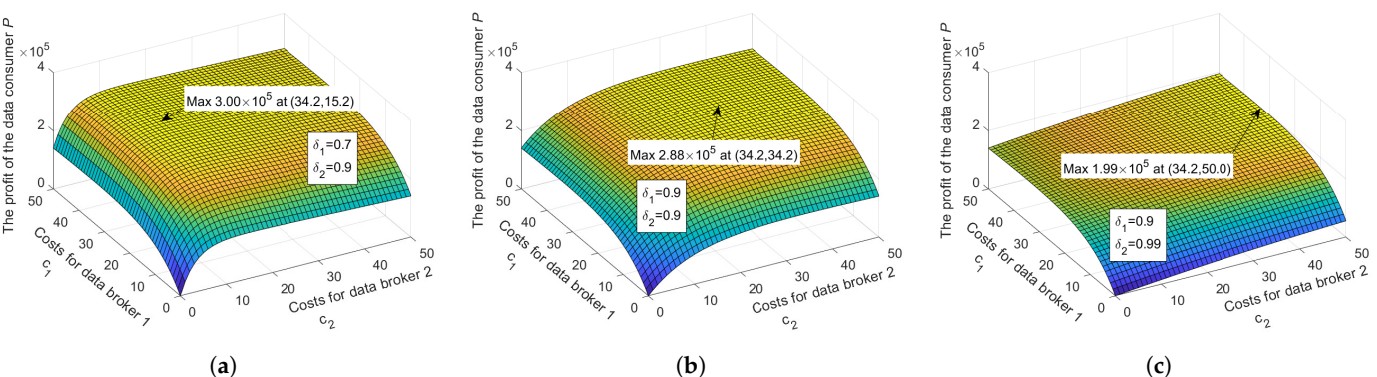

**Figure 4.** The trends of the profit for data consumers with the two data brokers ($n_1 = n_2 = 500$ and $U = 1000$. (**a**) $\delta_1 = 0.9$ and $\delta_2 = 0.7$. (**b**) $\delta_1 = 0.9$ and $\delta_2 = 0.9$. (**c**) $\delta_1 = 0.9$ and $\delta_2 = 0.99$.)

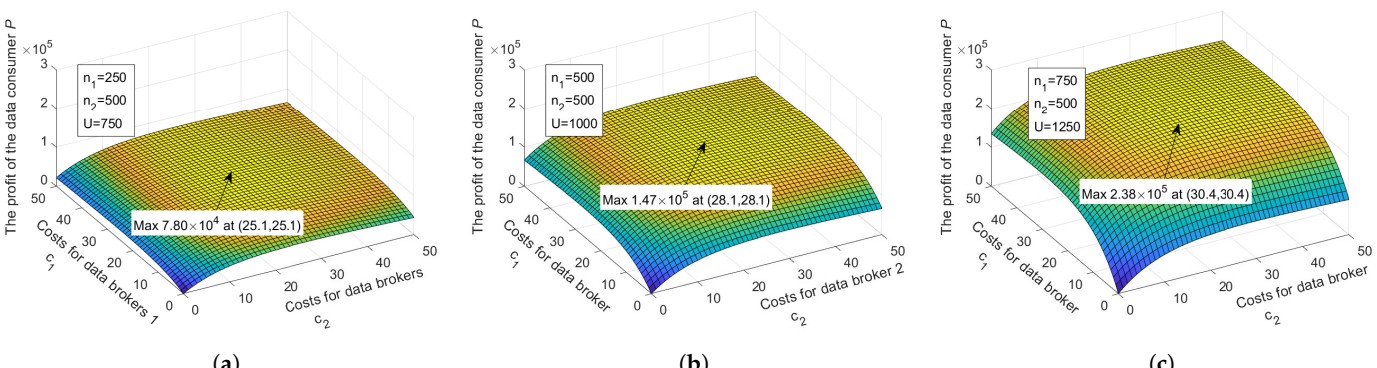

**Figure 5.** The trends of the profit for data consumers with the two data brokers ($\delta_1 = \delta_2 = 0.9$. (**a**) $n_1 = 250$, $n_2 = 500$, $U = 750$. (**b**) $n_1 = 500$, $n_2 = 500$, $U = 1000$. (**c**) $n_1 = 750$, $n_2 = 500$, $U = 1250$).

*5.2. Experiments with Real-World Parameters*

For privacy sensitivity factor $\delta$ used to analyze the proposed WTS function, this paper introduces five personal data types based on the real-world survey in [29] that investigated the opinion of about 1000 people about the average price of their personal data. The survey asked respondents who would be willing to share their personal information in exchange for money to indicate what would be the minimum amount they would accept as compensation in exchange for different data categories.

Similar to the previous section, to set the privacy sensitivity parameter values, this paper assumes that 50% of the data providers participate in the market with the average price (i.e., $\Phi(\cdot) = 0.5$). Then, the privacy sensitivity parameters are set by 0.733, 0.883, 0.913, 0.946, and 0.966 (i.e., name ($3.9), phone number ($5.9), email address ($8.0), home address ($12.9), purchase histories ($20.6), which are generally used for marketing, target advertising, etc.), respectively. It means that there are five data brokers in this experiment.

In this experiment, it is checked how the boundary conditions (Equations (24) and (25)) affect the profit of the data consumer. In other words, the gap analysis between theoretical and practical approaches by checking the performance of the **Problem 3** with and without the boundary conditions, respectively.

Table 1 shows results of the proposed constrained profit maximization (**Problem 3**) without and with the proposed boundary conditions (24) and (25). This table consists of three different cases with respect to the amount of budget: (1) sufficient, (2) tight, and (3) inadequate cases. Meanwhile, the left-hand side values show the results without the boundary condition (24) representing theoretical analysis, and the right-hand side values show the results with the boundary condition (24) representing practical analysis.

**Table 1.** A results of profit maximization of the data consumer for **Problem 3**.

| | | 5 data brokers, $n_i = 500$, $\bar{\delta} = (0.733, 0.883, 0.913, 0.946, 0.966)$, $\alpha = 0.5$ | | | | |
|---|---|---|---|---|---|---|
| | | **Problem 3** without boundary condition (24) | | **Problem 3** with boundary condition (24) | | |
| **Case 1:** Sufficient budget $B = 200,000$ | $\bar{c}$ | $(17.9, 33.6, 41.6, 57.8, 77.6)$ | $P = 1,002,123$ | $\bar{c}$ | $(8.41, 18.0, 23.9, 38.0, 59.7)$ | $P = 945,146$ |
| | WTS (%) | $(99.2, 98.0, 97.3, 95.6, 92.9)$ | | WTS (%) | $(89.4, 87.8, 87.5, 87.1, 86.9)$ | |
| | $\bar{b}$ | $(8.87 \times 10^3, 1.65 \times 10^4,$ $2.03 \times 10^4, 2.76 \times 10^4, 3.60 \times 10^4)$ | $\sum_i b_i = 109,262$ | $\bar{b}$ | $(3.76 \times 10^3, 7.91 \times 10^3,$ $1.05 \times 10^4, 1.65 \times 10^4, 2.60 \times 10^4)$ | $\sum_i b_i = 64,619$ |
| **Case 2:** Tight budget $B = 50,000$ | $\bar{c}$ | $(12.2, 20.4, 23.9, 29.9, 35.5)$ | $P = 927,297$ | $\bar{c}$ | $(8.41, 18.0, 23.9, 32.5, 39.2)$ | $P = 919,124$ |
| | WTS (%) | $(96.1, 90.8, 87.5, 80.1, 70.1)$ | | WTS (%) | $(89.4, 87.8, 87.5, 82.7, 73.6)$ | |
| | $\bar{b}$ | $(5.84 \times 10^3, 9.25 \times 10^3,$ $1.05 \times 10^4, 1.20 \times 10^4, 1.25 \times 10^4)$ | $\sum_i b_i = 50,000$ | $\bar{b}$ | $(3.76 \times 10^3, 7.91 \times 10^3,$ $1.05 \times 10^4, 1.35 \times 10^4, 1.44 \times 10^4)$ | $\sum_i b_i = 50,000$ |
| **Case 3:** Inadequate budget $B = 10,000$ | $\bar{c}$ | $(5.99, 7.86, 8.40, 9.07, 9.52)$ | $P = 584,408$ | $\bar{c}$ | $(5.99, 7.86, 8.40, 9.07, 9.52)$ | $P = 584,408$ |
| | WTS (%) | $(79.8, 60.1, 51.8, 38.7, 27.7)$ | | WTS (%) | $(79.8, 60.1, 51.8, 38.7, 27.7)$ | |
| | $\bar{b}$ | $(2.39 \times 10^3, 2.36 \times 10^3,$ $2.18 \times 10^3, 1.75 \times 10^3, 1.32 \times 10^3)$ | $\sum_i b_i = 10,000$ | $\bar{b}$ | $(2.39 \times 10^3, 2.36 \times 10^3,$ $2.18 \times 10^3, 1.75 \times 10^3, 1.32 \times 10^3)$ | $\sum_i b_i = 10,000$ |

For the sufficient budget case (Case 1), the profit values of the data consumer are significantly different. Particularly, the results without the boundary condition spend much budget to earn more profit. However, as shown in the WTS column, the portion of data providers is more than 90% (from 92% to 99%), which is relevantly extreme. On the other hand, the results with the boundary condition spend less budget due to cost boundary and show a more feasible WTS range (from 86% to 89%). Note that the sufficient case represents that the data consumer has enough budget, so it can ignore the budget constraint condition (condition (25)).

With the tight budget case (Case 2), for both cases with/without boundary conditions, the profit values of the data consumer are similar, and the total amount of spend budget ($\sum_i b_i$) are the same. However, the detailed results such as cost allocation ($\bar{c}$) and the portion of data providers (WTS) are quite different. Without the boundary condition, the data

consumer can allocate more costs to buy less privacy-sensitive data (i.e., a lower value of $\delta$) to increase the size of the dataset. However, with the boundary condition, the data consumer has a limitation in allocating costs to a certain data broker. Therefore, for $\delta$ of 0.733, the former can allocate a cost of 17.9 for obtaining WTS of 96.1%, but the latter can allocate a cost of 8.41 for obtaining WTS of 89.4%. Meanwhile, the latter case can allocate more cost to buy more privacy-sensitive data (i.e., a higher value of $\delta$). In other words, for $\delta$ of 0.996, the former can allocate a cost of 35.5 for obtaining WTS of 70.1%, but the latter can allocate a cost of 39.2 for obtaining WTS of 73.6%. Therefore, the proposed boundary condition (condition (24)) can force to buy personal data from more variety of data brokers.

With the inadequate budget case (Case 3), both cases with/without boundary conditions show identical results in terms of cost allocation ($\bar{c}$) and the portion of data providers (WTS), budget allocation ($\bar{b}$), the amount of spend budget ($\sum_i b_i$), the profit of the data consumer ($P$). It shows that if there is only a small amount of budget available, there is no chance of meeting the boundary condition while performing the profit maximization problem.

So far, it has been observed that a single point of profit maximization of the data consumer. From now on, this paper observes how the profit values of the data consumer change. For this experiment, this paper assumes each data broker has $\delta_i$ in the range of $[0.3, 0.99]$ (i.e., from \$1 to \$90 for the average costs) under the uniform random distribution, and it simulates 1000 times to have the average values of profit. Similar to the other experiments, this paper sets the value of $\alpha$ as 0.5 and observes the proposed constrained profit maximization problem (**Problem 3**) with enough budgets.

Figure 6 shows the trends of profit values of the data consumer with respect to the various number of data providers and service users with multiple data brokers. For one case in Figure 6a, the number of service users is fixed to 1000, and the number of data providers for each data broker is changed from 100 to 250. When the number of data providers increases, the profit of the data consumer also increases because the data consumer can utilize high-quality datasets to increase the service quality for the service users. In other words, the number of paid service users increases while the number of data providers increases. Therefore, when the number of data brokers increases, the profit of the data consumer rapidly increases because adding a data broker means multiplying data providers.

For the other case in Figure 6b, the number of data providers for each data broker is set to 200, and the number of service users is changed from 800 to 1200. Similar to the previous case, when the number of service users increases, the profit of the data consumer increases. However, the slope of this case is more gradual than that of the previous case.

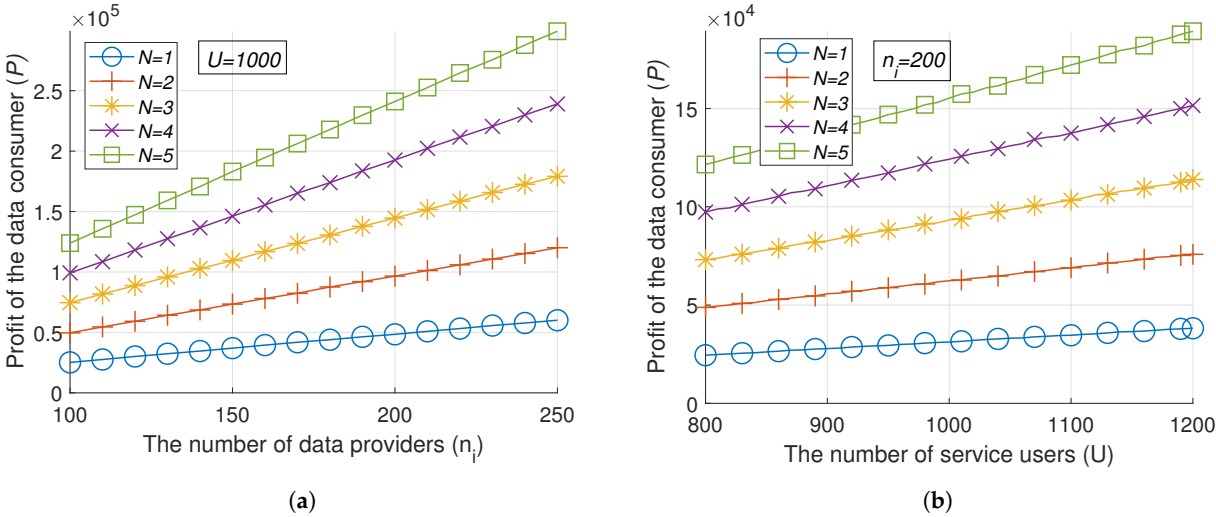

(**a**) (**b**)

**Figure 6.** The trends of the profit for data consumers with the multiple data brokers. (**a**) Various number of data providers. (**b**) Various number of service users.

From the data consumer's perspective, to increase its profit, finding more data providers as well as service users is important. Particularly, by securing more data providers, the data consumer has more chance to create a more high-quality service for the service users, which makes the service users actually pay for the service to increase the profit of the data consumer.

*5.3. Discussions*

Since this paper proposes a profit maximization model for the data consumers in a consent-based personal data trading model and analyzes various performances in theoretical and practical aspects, there are several discussion points.

5.3.1. The Correlation among $\delta$, $\alpha$, and Data Quality

In this paper, the correlation between the privacy sensitivity factor $\delta$ and a service quality factor $\alpha$ is not considered. In general, personal data with larger privacy sensitivity are likely to have more quality information that directly affects the service quality of the data consumer. Also, the data quality model itself should be considered. In this paper, only data size is considered to increase the service quality of the data consumer. However, since the data consumer collects various types of personal data at the same time, it is possible to achieve additional information from the dataset by combining various personal data types. Therefore, it is important to discover the correlation among the privacy sensitivity factor, the data quality of the gathered personal dataset, and the expected service quality of the data consumer in terms of setting parameters. Several studies, including [25,35], have tackled measuring and/or modeling the correlations, but still it is necessary to further studies on this topic.

5.3.2. Time Complexity of SLSQP

This paper chooses the sequential least squares programming (SLSQP) method in SciPy to solve the proposed **Problem 3**, which is a constrained nonlinear optimization problem. The SLSQP is one of the trust-region square programming methods, and it is hard to explicitly calculate the time complexity of these kinds of solvers in Big-O notation. However, fortunately, Varelas and Dahito [36] performed benchmark tests about many multivariate solvers of SciPy on the noiseless testbed, and the SLSQP solver has the best performance in terms of the average runtime and the statistical significance.

**6. Conclusions**

This paper has proposed a profit maximization model for data consumers with data providers' incentives in the personal data trading market. By jointly considering the behavioral models of personal data providers and service users in terms of the data consumer, this paper has proposed a constrained profit maximization problem with a limited budget for the data consumer. By analyzing numerical results in theoretical and practical aspects, this paper has shown the feasibility of the proposed profit maximization model. The proposed model implies that the personal data market can be formed even if the data consumer spends direct incentives to data providers as costs. Since direct incentives can encourage data providers' market participation, this model can help to wide data trading ecosystem. In the future, it should be considered that more realistic models for data providers (i.e., willingness-to-sell personal data) and service users (i.e., willingness-to-pay offered service) as well as the correlation among various parameters about data quality and service quality.

**Author Contributions:** Conceptualization, H.P. and H.O.; methodology, H.P. and H.O.; validation, H.P., H.O., and J.K.C.; formal analysis, H.P. and H.O.; investigation, H.P. and H.O.; writing—original draft preparation, H.P. and H.O.; writing—review and editing, H.O. and J.K.C.; visualization, H.P. and H.O.; supervision, H.O.; project administration, H.O.; funding acquisition, H.O. All authors have read and agreed to the published version of the manuscript.

**Funding:** This work was supported by the National Research Foundation of Korea (NRF) grant funded by the Korea government (MSIT) (NRF-2022R1C1C2003437).

**Institutional Review Board Statement:** Not applicable.

**Informed Consent Statement:** Not applicable.

**Data Availability Statement:** No new data were created or analyzed in this study. Data sharing is not applicable to this article.

**Acknowledgments:** The authors would like to thank anonymous reviewers for taking the time and effort necessary to review the manuscript and providing all valuable comments and suggestions to improve the quality of the manuscript.

**Conflicts of Interest:** Author Hyojin Park was employed by the company TOVDATA Inc. The remaining authors declare that the research was conducted in the absence of any commercial or financial relationships that could be construed as a potential conflict of interest.

## Abbreviations

The following abbreviations are used in this manuscript:

| Symbol | Definition |
|---|---|
| $N$ | Number of data brokers in the market |
| $n_i$ | Number of potential data providers with data broker $i$ |
| $\bar{\delta}$ | Privacy sensitivity of data providers $\{\delta_1, \delta_2, ..., \delta_i\}$ |
| $\Phi$ | Data provider's willingness-to-sell function |
| $B$ | The entire budget of the data consumer |
| $\bar{b}$ | budget allocation for data brokers $\{b_1, b_2, ..., b_i\}$ |
| $\bar{c}$ | cost allocation for each data provider $\{c_1, c_2, ..., c_i\}$ |
| $U$ | Number of potential service users |
| $s$ | Service fee paid by each service user |
| $\Psi$ | Service user's willingness-to-pay function |
| $P$ | Profit function of the data consumer |
| $Q$ | Service quality function of the data consumer |
| $D$ | Size of the entire dataset collected by all data brokers |
| $d_i$ | Size of dataset collected by data broker $i$ |

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
