# Peer review of "A Profit Maximization Model for Data Consumers with Data Providers’ Incentives in Personal Data Trading Market"

_data_

Round 1
Reviewer 1 Report
Comments and Suggestions for Authors
Dear authors, I have really enjoyed reading your manuscript, it has a lot of potential to be published, as long as you can consider the following observations:
1. The introduction of the manuscript is well nourished, however it lacks aspects related to the dimensionality of the data, geographical and social issues of consumers and suppliers.
2. Regarding the related works of the state of the art, in line 126, it is important to justify the use of IoT-based models, in the same way in line 144.
3. Regarding the quality of service, in lines 276 and 277, they propose accuracy as the main metric of the model, why?
Greeting

Reviewer 2 Report
Comments and Suggestions for Authors
Structure and organization of the paper can be improved for better clarity. It lacks a clear introduction, background, and problem statement, making it difficult for readers to understand the context and significance of the research.
The paper mentions that "many countries have established new data regulations," but it doesn't provide specific references or details about these regulations. Providing examples or references to these regulations would enhance the paper's credibility and context.
The paper discusses data brokers collecting personal data with or without explicit consent. It would be good to discuss the ethical implications of this and how the proposed profit maximization model addresses these ethical concerns.
The paper mentions using parameters inspired by real-world surveys on data providers, but it lacks information on the methodology, data sources, or the size and representativeness of the survey. More comprehensive details are required.
Illustrative examples need better presentation and detailing in problem statement.
The paper mentions considering the behavior of data providers and service users under a limited budget, but it doesn't explain how this budget constraint is integrated into the model or its impact on the results.
The paper introduces the concept of a "behavioral model of a data consumer" but doesn't provide a detailed explanation of this model. Clarifying how data consumer behavior is modeled and its relevance to the research would be helpful.
Average
Reviewer 3 Report
Comments and Suggestions for Authors
I congratulate the authors for this interesting research. however, I have few comments that might help to improve it;
The abstract should start with the research objective "This paper proposes a profit maximization model for a data consumer when it buys personal data from data providers (by obtaining consent) through data brokers and provides their new services to data providers". the focus of the abstract should be the findings/indication rather than the background of the study
In the related work, the authors could consider discussing the limitations of the existing models in more detail, to better position their contribution.
also, the authors have provided a list of several model developed by scholars in describtive manner. for example, "Mao et al. [28] proposed a double...", "Pal et al. [29] proposed a preference-based...." "Li et al. [30] proposed an IoT...." then what ?? such discussion should have a point to be proven !!
in the results, Include a subsection that directly compares these results with existing models or real-world scenarios to highlight the model's applicability.
discuss the implication of the study in the conclusion and recommendations for further research.
Comments on the Quality of English Languageproofreading is needed. for example, "...organized as follows. Section 2 shows" should be "...organized as follows; Section 2 shows"
Round 2
Reviewer 3 Report
Comments and Suggestions for Authors
I am pleased to see the modified version and the improved quality.
Good luck
Author Response
We appreciate your valuable comments for this manuscript